

# Model-measurement comparison of functional group abundance in α-pinene and 1,3,5-trimethylbenzene secondary organic aerosol formation

G. Ruggeri[1], F. A. Bernhard[1], B. H. Henderson[2], and S. Takahama[1]

[1]ENAC/IIE Swiss Federal Institute of Technology Lausanne (EPFL), Lausanne, Switzerland
[2]Department of Environmental Engineering Sciences, University of Florida, Gainesville, FL, USA

*Correspondence to:* Satoshi Takahama (satoshi.takahama@epfl.ch)

**Abstract.** Secondary organic aerosol (SOA) formed by α-pinene and 1,3,5-trimethylbenzene photooxidation under different $NO_x$ regimes is simulated using the Master Chemical Mechanism v3.2 (MCM) coupled with an absorptive gas/particle partitioning module. Vapor pressures for individual compounds are estimated with the SIMPOL.1 group contribution model for determining apportionment of reaction products to each phase. We apply chemoinformatic tools to harvest functional group

(FG) composition from the simulations and estimate their contributions to the overall oxygen to carbon ratio. Furthermore, we compare FG abundances in simulated SOA to measurements of FGs reported in previous chamber studies using Fourier Transform Infrared Spectroscopy. These simulations qualitatively capture the dynamics of FG composition of SOA formed from both α-pinene and 1,3,5-trimethylbenzene in low $NO_x$ conditions, especially in the first hours after start of photooxidation. Higher discrepancies are found after several hours of simulation; the nature of these discrepancies indicate sources of uncertainty or

types of reactions in the condensed or gas phase missing from current model implementation. Higher discrepancies are found in the case of α-pinene photooxidation under different $NO_x$ concentration regimes, which are reasoned through the domination by a few polyfunctional compounds that disproportionately impact the simulated FG abundance in the aerosol phase. This manuscript illustrates the usefulness of FG analysis to complement existing methods for model-measurement evaluation.

## 1 Introduction

Atmospheric aerosols are complex mixtures that can contain a multitude of chemical species (Seinfeld and Pandis, 2006). While the inorganic fraction comprises a relatively small number of compounds, the organic fraction (or organic aerosol, OA) includes thousands of compounds with diverse molecular structures (Hamilton et al., 2004). These compounds take part in multitude of gas phase, aerosol phase, and heterogeneous transformation processes (e.g., Kroll and Seinfeld, 2008; Hallquist et al., 2009; Ziemann and Atkinson, 2012) that must be modeled with sufficient fidelity to predict atmospheric concentrations

and impacts from various emission scenarios.

    A mechanism central to these processes is the formation of semivolatile organic compounds (SVOCs) through gas-phase oxidation of volatile organic compounds (VOC) precursors and their reaction products. α-pinene (APIN) and 1,3,5-trimethylbenzene (TMB) are examples of biogenic and anthropogenic VOC precursors, respectively, which have been studied for their chemical





reaction mechanisms and aerosol yields in environmentally-controlled chamber experiments and numerical simulation. APIN is a monoterpene compound primarily emitted from coniferous vegetation (Fuentes et al., 2000; Tanaka et al., 2012) with high emission rate, reactivity, and secondary organic aerosol (SOA) generation potential (e.g., Fehsenfeld et al., 1992; Lamb et al., 1993; Chameides et al., 1988; Jenkin, 2004; Tolocka et al., 2004; Sindelarova et al., 2014). TMB is an aromatic compound

emitted from vehicular emissions and a major contributor to urban organic aerosol (e.g., Kalberer et al., 2004); its degradation mechanism has also been subject of collective evaluation (Metzger et al., 2008; Wyche et al., 2009; Rickard et al., 2010; Im et al., 2014). Gas-phase oxidation reactions are modeled with chemically explicit or semi-explicit treatment, or alternatively using a basis set approach based on simplified molecular or property descriptors; SOA formation is commonly modeled by coupling these reactions with partitioning of oxidation products to an absorptive organic phase (e.g., Jenkin et al., 1997; Pun

et al., 2002; Griffin et al., 2003; Aumont et al., 2005; Capouet et al., 2008; McFiggans et al., 2010; Barley et al., 2011; Chen et al., 2011; Murphy et al., 2011; Aumont et al., 2012; Jathar et al., 2015). SVOCs produced by such reactions can in reality partition among multiple phases (vapor, organic liquid, aqueous, solid), and participate in additional functionalization, accretion, or fragmentation reactions in one of many phases (Kroll et al., 2011; Cappa and Wilson, 2012; Im et al., 2014; Zhang and Seinfeld, 2013; Zhang et al., 2015). These processes are represented in models with varying degrees of detail; simplifying

or wholly omitting various mechanisms out of concerns for computational feasibility or lack of sufficient knowledge. For instance, in a work we follow closely in this manuscript, Chen et al. (2011) used a fully explicit gas-phase reaction mechanism with absorptive organic partitioning and evaluated the potential importance of missing heterogeneous and condensed-phase mechanisms based on discrepancy of model simulation and experiments.

Our capability to simulate SOA formation is often evaluated against aerosol mass yield, O:C, carbon oxidation state, mean

carbon number, volatility, and specific species or compound classes when available (e.g., Robinson et al., 2007; Kroll et al., 2011; Donahue et al., 2012; Nozière et al., 2015). These properties can be measured using various forms of mass spectrometry (e.g., Jayne et al., 2000; Jimenez et al., 2009; Nizkorodov et al., 2011); or monitoring changes in size distribution in combination with isothermal dilution or thermal heating (e.g., Grieshop et al., 2009; Cappa, 2010; Epstein and Donahue, 2010; Donahue et al., 2012). Functional group (FG) composition is a complementary representation of organic molecules and complex organic

mixtures that offers a balance between parsimony and chemical fidelity for measurement and interpretation.

FGs represent structural units of molecules that play a central role in chemical transformations, and provide insight into evolution of complex organic mixtures without monitoring all species explicitly (Holes et al., 1997; Sax et al., 2005; Presto et al., 2005; Lee and Chan, 2007; Chhabra et al., 2011; Zeng et al., 2013). FG abundances have also been associated with volatility (e.g., Pankow and Asher, 2008), hygroscopicity (e.g., Hemming and Seinfeld, 2001; Suda et al., 2014), and magnitude

of nonideal interactions in the condensed phase (e.g., Ming and Russell, 2002; Griffin et al., 2003; Zuend et al., 2011). However, two impediments have been the likely cause of slow adoption of this representation. Building quantitative calibration models of FG abundance have posed analytical challenges, but rapid progress has been made over the past decade with Fourier Transform Infrared Spectroscopy (FTIR) (e.g., Sax et al., 2005; Reff et al., 2007; Coury and Dillner, 2008; Day et al., 2010; Takahama et al., 2013; Ruthenburg et al., 2014; Takahama and Dillner, 2015), Nuclear Magnetic Resonance (Decesari et al., 2007;

Cleveland et al., 2012), spectrophotometry (Aimanant and Ziemann, 2013), and GC-MS with derivatization (Dron et al., 2010).





The second challenge is computationally harvesting FG abundance from a large set of known molecular structures. To this end, Ruggeri and Takahama (2015) developed a set of substructure definitions corresponding to FGs that can be queried against arbitrary molecules specified by their molecular graphs.

In this work, we apply these new substructure definitions to describe the FG composition of products simulated by gas-phase reactions prescribed with the Master Chemical Mechanism (MCMv3.2) (Jenkin et al., 1997; Saunders et al., 2003; Jenkin et al., 2003; Bloss et al., 2005), and SOA constituents formed by their dynamic absorptive partitioning (Chen et al., 2011). Three instances of APIN photooxidation under varying initial concentrations of oxides of nitrogen ($NO_x$), and TMB oxidation in the presence of $NO_x$ are studied in accordance to aerosol FG composition characterized by Sax et al. (2005) and Chhabra et al. (2011) in chamber studies using FTIR. The model results are analyzed through a suite of FG abundances and model-measurement comparisons of measured FGs are presented to hypothesize reasons (including unimplemented mechanisms) for discrepancies where they occur.

## 2 Methods

### 2.1 Systems studied

Photooxidation of APIN under "low $NO_x$" ($NO_x$/APIN of 0.8), "high $NO_x$" ($NO_x$/APIN of 18), and no $NO_x$ conditions (designated as $lNO_x$, $hNO_x$, and $nNO_x$, respectively), and TMB under "low $NO_x$" ($NO_x$/TMB ratio of 0.24; designated as $lNO_x$) conditions were simulated in this study to compare with available measurements of aerosol FG composition in environmental chamber experiments. Simulations were run at 298 K and conditions closely following experimental descriptions summarized in Table 1, with a few exceptions. In the case of APIN degradation in high $NO_x$ conditions, the $H_2O_2$ was used as the OH radical initiator as $CH_3ONO$ is not available in the MCMv3.2 degradation scheme. When the reacted instead of initial precursor concentration is reported, this value is used as the initial concentration for the simulations. This decision is supported by the virtual observation that 99% of the precursor is reacted after 4.5–6.5 hours in these cases (Figure S1), and specification of higher initial concentrations lead to reacted quantities inconsistent with experimental specifications.

### 2.2 Model formulation

While differing in implementation, the model specification resembles the MCM-SIMPOL framework described by Chen et al. (2011). The chemical mechanism prescribed by MCMv3.2 (Jenkin et al., 1997; Saunders et al., 2003; Jenkin et al., 2003; Bloss et al., 2005) was used to simulate the gas-phase oxidation of volatile organic compounds (VOCs). The Kinetic Pre-Processor (KPP, Damian et al., 2002; Sandu and Sander, 2006; Henderson, 2016) was used to generate the gas-phase chemistry code in Fortran 90. A separate dynamic absorptive partitioning (Pankow, 1994) module was added via sequential operator splitting (Yanenko, 1971; Orlan and Boris, 2000; Vayenas et al., 2005) to simulate gas/particle (G/P) partitioning after the reaction operator. Pure component vapor pressures of organic compounds in the MCMv3.2 degradation schemes were calculated using SIMPOL.1 (Pankow and Asher, 2008), and non-ideal interactions were neglected in these simulations (i.e., activity coefficients





were set to unity for all species). Vapor pressures are converted to equivalent mass concentrations $C^0$ (Appendix A), and normalized by a reference value for presentation in logarithmic units (Seinfeld and Pandis, 2006) such that the notation $\log_{10} C^0$ implies $\log_{10}(C^0/1\,\mu g\,m^{-3})$. LSODE (Livermore Solver for Ordinary Differential Equations; Radhakrishnan and Hindmarsh, 1993) was used as the numerical solver for each operation (reaction and G/P partitioning). A time step of 60 seconds is used in this study, as it is in the order of magnitude of the timescale of gas-phase oxidation and condensation/evaporation under chamber conditions (Cocker et al., 2001) and leads to stable solutions. Radiation intensities were fixed to maximum values corresponding to clear sky conditions at an altitude of 0.5 km, 1° solar zenith angle in July, and a latitude of 45 N (Derwent et al., 1996; Hayman, 1997; Derwent et al., 1998; Saunders et al., 2003).

Absorptive partitioning to a purely organic phase is considered in this model (Appendix A in Supplemental Material). The relative humidity (RH) specified in the experiments are converted to equivalent concentrations of $H_2O$ for participation in the $HO_2$ radical self reaction to form hydrogen peroxide (Mozurkewich and Benson, 1985), but water uptake by the aerosol and its influence on G/P partitioning of organic compounds (Seinfeld et al., 2001; Chang and Pankow, 2010) is not considered. As aerosol growth following homogeneous and heterogeneous nucleation processes of the condensed organic phase in the chamber experiments are not included in the model, we use a seed $C_{OA,init}$ of 1 $\mu g\,m^{-3}$ to initiate G/P partitioning (Appendix B). The relative measures reported in this study are insensitive to this value (Figures S2–S4). To differentiate between the SOA formed in the simulation and the total organic aerosol phase involved in partitioning, we denote the former quantity as $C_{SOA}$ the latter as $C_{OA} = C_{OA,init} + C_{SOA}$. No condensed-phase reactions are included; as with Chen et al. (2011) we consider them as a potential source of model-measurement discrepancies. While the particle diameter of the monodisperse population is allowed to grow according to the organic aerosol condensed (Section B), the number concentration of particles is kept fixed during the simulation; losses of both particles and gases to chamber walls (e.g., Loza et al., 2010; Matsunaga and Ziemann, 2010; Zhang et al., 2014) are neglected. These assumptions will affect calculations of total yield and rate of change in aerosol mass; however, aerosol mass yields are in the range of physical expectation (Figure S5; mass concentrations represented in the volatility basis set convention are also shown in Figure S6 for reference). Relative abundances of functional groups are robust with respect to many of these assumptions and will be the primary focus of our presentation and model-measurement comparisons.

## 2.3 Simulation analysis

A chemoinformatic tool (APRL-SSP; Takahama, 2015) described by Ruggeri and Takahama (2015) is used to harvest FG abundances (enumeration of the FG fragments) from each molecule in the simulations. This tool consists of scripts invoking OpenBabel and Pybel (O'Boyle et al., 2008, 2011) and SMARTS patterns (DAYLIGHT Chemical Information Systems, Accessed 30 September 2015) formulated and validated for these chemical systems. Using this tool, molecular structure is mapped to input parameters for SIMPOL.1, and FG abundances of the organic aerosol mixture are obtained from molecular concentrations. Most importantly, we extract two arrays with elements $\phi_{ip}$, the number of times FG $p$ occurs in molecule $i$, and $\phi_{ipa}^{*}$, the number of times atom type $a$ occurs in FG $p$ in molecule $i$. We combine these two coefficients with the molecular or molar concentrations $C$ of compound $i$ in phase $\alpha$ generated by our simulations to estimate several useful mixture properties





for time $t_j$:

$$\sum_{i \in \mathcal{M}} C_i^{\alpha}(t_j)\, \phi_{ip} = \text{abundance of FG } p$$

$$C_i^{\alpha}(t_j)\, \phi_{ip} / \left( \sum_{i \in \mathcal{M}} C_i^{\alpha}(t_j)\, \phi_{ip} \right) = \text{fractional contribution of molecule } i \text{ to abundance of FG } p$$

$$\sum_{i \in \mathcal{M}} C_i^{\alpha}(t_j)\, \phi_{ipa}^* = \text{apportionment of atoms of type } a \text{ to FG } p \, .$$

The summation is taken for the set of all compounds (or molecule types) $\mathcal{M}$. The last quantity is used to separate the contributions of O:C and N:C from various FGs. The set of patterns were constructed to meet conditions of completeness and specificity (each atom is matched only by one and only one group) such that the sum of oxygen and nitrogen atoms in each FG sums to the total number of atoms in the system (Ruggeri and Takahama, 2015). Polyfunctional carbon atoms are not considered in the condition for specificity — matches by multiple groups leads to overestimation of counts in $\phi_{ipa}^*$ — therefore, the total number of carbon used in the denominator of these atomic ratios is estimated using the SMARTS pattern `[#6]`.

We additionally estimate Integrated Reaction Rates (IRR; Jeffries and Tonnesen, 1994) to examine degradation rates relative to rates of production in the gas phase (g) for selected systems. IRR for reaction $r$ affecting compound $i$ at time $t_j$ is calculated from the rate constant $k$ and the product of concentrations $C$:

$$IRR_{ri}(t_j) = C_i^{(g)}(t_j) - C_i^{(g)}(t_j - \Delta t) = \int_{t_j - \Delta t}^{t_j} dt \left( k_r \prod_{i' \in \mathcal{M}_r} C_{i'}^{(g)}(t) \right)$$

$$\approx \Delta t \left( k_r \prod_{i' \in \mathcal{M}_r} C_{i'}^{(g)}(t_j) \right) .$$

$\mathcal{M}_r$ is the set of compounds involved in reaction $r$. The expression in parentheses is the conventional rate equation for reaction $r$. To obtain the IRR for functional group $p$, we multiply by the factor $\phi_{ip}$ described above:

$$IRR_{rp}(t_j) = \sum_{i \in \mathcal{M}_r} IRR_{ri}(t_j)\, \phi_{ip} .$$

IRR estimates were harvested from the LSODE solver, and the PERMM package (Henderson, 2015) used to associate compounds and FGs with each reaction.

## 2.4 Measurements

FTIR analysis reported by Sax et al. (2005) and Chhabra et al. (2011) quantified the molar abundance of alkane CH (aCH), carboxylic acid (COOH), non-acid (ketone and aldehyde) carbonyl (naCO), alcohol OH (aCOH), and organonitrate (CONO$_2$) FGs. FTIR analysis has been found to measure around 80% of the organic mass (OM) in environmental or laboratory generated OA (Maria et al., 2002). Uncertainties in the FG quantification have been reported to be between 5 and 30% (Russell, 2003; Takahama et al., 2013). Sax et al. (2005) collect particles in the range of 86-343 nm onto zinc selenide substrates by



impaction, while Chhabra et al. (2011) sample generated aerosol onto Polytetrafluoroethylene (PTFE) filters for FTIR analysis. Measurement artifacts can arise during time-integrated collection of aerosol samples and can differ according to duration of sampling (Subramanian et al., 2004) or method of collection (Zhang and McMurry, 1987). The primary driver for absorptive and evaporative artifacts which may impact bulk mass estimation is the difference between the changing gas-phase composi-

tion and equilibrium vapor composition with respect to the aerosol phase, but model simulations suggest the relative gas-phase composition stabilizes after the first few hours. Changes in particle composition due to condensed-phase chemistry may perturb the equilibrium, but this phenomenon may be interpreted together with condensed-phase processes not included in the model. In the analysis of Chhabra et al. (2011), samples transported off-site for analysis were frozen to minimize evaporation and reaction artifacts during storage. Additionally, evaporative losses in the analysis chamber of the FTIR (during purging of

headspace with dry nitrogen gas) were minimized by rapid scanning, and Sax et al. (2005) report that the spectrum was stable even when repetitive measurements are performed.

In this work, we limit our discussion to results based on molar rather than mass concentrations of FG abundances. While mass concentrations are commonly reported for FTIR measurements of ambient samples (e.g., Russell et al., 2009), estimates are based on fixed assumptions regarding the apportionment of polyfunctional carbon atoms to associated FGs (e.g., Allen

et al., 1994; Russell, 2003; Reff et al., 2007; Takahama et al., 2013; Ruthenburg et al., 2014). These assumptions can affect both mass estimation and atomic ratios (e.g., O:C). Chhabra et al. (2011) proposed a modification based on assumed molecular structures in their chamber experiments, and mass estimates using these values are shown in Figure S7. Constraining the mapping of measured bonds to atoms for estimation of these quantities in various mixtures are planned for future work.

## 3   Results and Discussion

In each of the following sections, we begin by describing the simulated evolution of FGs primarily in terms of their contribution to the O:C ratio (Figure 1), and then discuss comparisons of mole fractions with observations for a subset of measured FGs (Figures 2 and 3).

### 3.1   APIN-lNO$_x$

#### 3.1.1   Simulation results

Initially, only the most oxygenated species condense to the aerosol phase, but oxygenated products continue to be formed in the gas phase and the O:C values exceeds the aerosol-phase O:C after four hours. The O:C ratios approach 0.75 and 0.6 for the gas and aerosol phases, respectively, after 20 hours of simulation (Figure 1). The O:C ratio of the simulated aerosol phase is comparable to the O:C ratio measured by Chen et al. (2011) and Zhang et al. (2015) in ozonolysis and photooxidation experiments without NO$_x$ ($\sim$0.5 in both cases).

The FG that contributes the most to the aerosol O:C ratio after 20 hours is hydroperoxide (31%), while in the gas phase the peroxyacyl nitrate is the major contributor (carrying five oxygen atoms per peroxyacyl nitrate FG) with 55% of the O:C





ratio of the gas phase mixture. Some peroxyacyl nitrates are also partitioned to the aerosol phase as reported in laboratory measurements (Jang and Kamens, 2001), but make a smaller contribution (12%) to the aerosol O:C. aCOH and $CONO_2$ FGs are found in higher abundance in the aerosol phase than many other FGs (Section S4) and contribute to the aerosol-phase O:C, while contributing negligibly to the gas-phase O:C. The large contribution of hydroperoxide FG to the aerosol-phase O:C is

consistent with their large contributions to SOA mass suggested in previous studies (Bonn et al., 2004; Wang et al., 2011; Mertes et al., 2012).

Addition of COOH lowers the pure component vapor pressure of a given molecule by four orders of magnitude (Kroll and Seinfeld, 2008; Pankow and Asher, 2008), but contribution to gas and aerosol phase O:C are approximately equal. In the gas-phase, CH3CO2H (formed from degradation of the peroxyacid radical compound CH3CO3) constitutes 60% of the COOH

fraction (Figure 4) at maximum $C_{SOA}$ (9.3 hours). The aldehyde and ketone CO lower the pure component vapor pressure of around one order of magnitude (Kroll and Seinfeld, 2008), but their contribution to O:C is greater than COOH in the aerosol phase on account of the higher abundance of carbonyl-containing compounds. More than 80% of the moles of carbonyl in both the gas and aerosol phases are associated with ketone rather than aldehyde CO (Figure S8).

We note the prevalence of several large polyfunctional compounds contributing to aerosol-phase. Their cumulative contribu-

tions to the total abundance varies over time (Figures S9 and S10); their contributions at peak $C_{SOA}$ are shown in Figure 4. Four compounds (C97OOH, C98OOH, C106OOH, and C719OOH) comprise 70% of the ketone and 80% of the hydroperoxide abundance. 811PAN contributes 50% of the peroxyacyl nitrate and also 45% of the COOH. Illustrations for these compounds are provided in Table 2.

### 3.1.2 Model-measurement comparison of FG mole fractions

Qualitative changes in the mole fractions of COOH, aCOH and $CONO_2$ FGs over the initial values reported by Sax et al. (2005) is well captured by the model (Figure 2). The magnitude of increase in COOH is higher in the measurements than in the model: an increase of 2.6 times against 1.5 times can be seen between the beginning and the end of the measurements and the simulation, respectively. For aCOH the discrepancy is smaller; an increase of 1.5 times from the beginning to the end of the experiments against 1.3 in the simulation is found. For $CONO_2$, the relative mole fraction decreases from 1 to 0.6 during

the experiment, while the model predicts a decrease to 0.3. For carbonyl (CO), the model is able to capture the general trend of initial decrease followed by an increase after 4 hours. The trend in modeled naCO is largely contributed by ketone, as it comprises more than 80% of the naCO (Figure S8). The magnitude of decrease in relative mole fraction of aCH observed by Sax et al. (2005) is not captured by the model. The measured relative mole fraction compared to the first sample decreases from 1 to 0.8, while its change is not detectable in the simulation (Figure 2).

The evolving differences in mole fractions between measurement and model are better viewed in Figure 3. The sum of the oxidized fraction in the simulation is consistently lower in the model predictions, remaining below 16% while increasing to 40% after 20 hours in the reported measurements. We consider two condensed-phase reaction mechanisms that may lead to such differences. Viewing the distribution of the compounds present in the MCM APIN-lNO$_x$ degradation scheme on $\log_{10} C^0$ vs. molar mass space (Figure 5), we see that the model does not include lower volatility compounds with molecular masses higher





than 300 $\mathrm{g\,mol^{-1}}$ observed in experiments (Shiraiwa et al., 2014). This high molecular mass fraction cannot entirely explain the missing COOH, aCOH, and naCO, however, as accretion reactions do not significantly increase the O:C of the mixture (Shiraiwa et al., 2014; Zhang et al., 2015). In the analysis by Shiraiwa et al. (2014), these compounds with high molecular mass and low volatility have an O:C ratio between 0.3 and 0.6. Furthermore, Zhang et al. (2015) report that around 60% of

the APIN SOA generated in environmental controlled chamber experiments is constituted by SVOCs, suggesting dimerization reactions can only partly be responsible for the discrepancies between simulations and experiments that we report in this study. Proposed dimerization reactions do not contribute to depletion of aCH bonds, and dimers produced in the aerosol phase have been found to have similar O:C ratio to the monomer (Zhang et al., 2015). Photolysis of hydroperoxides has been suggested as a condensed phase mechanism that leads to increase in naCO (Epstein et al., 2014), but an estimate based on the 6-day lifetime

molar conversion of hydroperoxide groups to naCO only increases the latter fraction from 8% to 9% of the FG mole fraction after 21 hours (though naCO increases by 9% over the case of no conversion), and does not fully explain the discrepancy between model and measurements for this FG. Further oxidation due to dissolved oxidants, such as OH radical, however may reduce the proportion of aCH relative to oxidized groups, though this rate is also dependent on diffusion and uptake of these radicals by the SOA (Donahue et al., 2013).

### 3.2  APIN-hNO$_x$

#### 3.2.1  Simulation results

While the FGs present in APIN-hNO$_x$ system are identical to the APIN-lNO$_x$ system, we find they occur in different proportions on account of both the lower ratio of VOC precursor to NO$_x$ concentrations, and lower absolute precursor concentrations. The predicted aerosol O:C ratio in this simulated system is approximately 0.75, while Chhabra et al. (2011) reports experimental

values around 0.4 according to AMS measurements. CONO$_2$ accounts for 47% of the simulated aerosol O:C after 20 hours (Figure 1). Both aldehyde and ketone CO contribute to O:C in the gas-phase more than in the aerosol phase, while CONO$_2$, aCOH, and COOH contribute primarily to O:C in the aerosol phase. The predicted aerosol N:C ratio is also overestimated ($\sim$0.1 in the simulated aerosol, Figure S11) compared to the measured value of 0.03, on account of the large contribution from CONO$_2$.

Lower precursor concentrations in the Caltech chamber experiments (Table 1) lead to lower concentration of condensible products in these corresponding simulations, enabling only a few compounds to partition to the aerosol phase in significant quantities (Figures S6, S9, and S10). The aerosol fraction of COOH exceeds 10%, but the rest remain below 5% of the gas phase, in contrast to the APIN-lNO$_x$ system where the aerosol fraction of six FGs exceed 10% (Figure S12). The aerosol mass yields on the order of a few percent (Figure S5) are consistent with $C_{\mathrm{SOA}}$ produced in the presence of high NO$_x$ concentrations

(e.g., Ng et al., 2007), where NO can compete for reaction with peroxy radicals that may otherwise produce lower volatility products. However, the overall $C_{\mathrm{SOA}}$ formed is an order of magnitude lower than the 54 $\mathrm{\mu g\,m^{-3}}$ reported in the experiments (Chhabra et al., 2011), which is surprising given that chamber experiments without wall loss corrections tend to underestimate true yields (e.g., Zhang et al., 2014). This underprediction may suggest the increasing importance of oligomer formation (e.g.,



Gao et al., 2004; Tolocka et al., 2004; Kalberer et al., 2006; Kroll and Seinfeld, 2008; Chen et al., 2011; Chhabra et al., 2011; Hall and Johnston, 2011) relative to the absorptive partitioning pathway at low $C_{OA}$ concentrations (Presto and Donahue, 2006). Sensitivity analyses conducted to increase the rate of condensation and overall $C_{SOA}$ formed had little impact on relative abundances estimated for FGs (Appendix B), so the interpretations presented are robust for the gas-phase reaction mechanisms

included and vapor pressures prescribed in our simulations.

In Figure 6, we see that C813NO3 is a polyfunctional compound that comprises 75% of $CONO_2$, 95% of COOH, and 70% ketone CO abundance, and 75% of the $C_{SOA}$ mass in the simulated aerosol at peak $C_{SOA}$ (3.2 hours). Polyfunctionality may introduce challenges in vapor pressure for linear group contribution methods such as SIMPOL.1, so we evaluate the uncertainty in vapor pressure prediction of the top five contributors (C813NO3, C98NO3, C719NO3, APINANO3, and APINBNO3) to

the $CONO_2$ abundance and $C_{SOA}$ mass by comparing to other methods (Table 3). SIMPOL.1 has been found to generally predict lower vapor pressures compared to other estimation methods like EVAPORATION (Compernolle et al., 2011) and the method of Nannoolal (Nannoolal et al., 2008), but in the case of mononitrates Compernolle et al. (2011) report that differences with EVAPORATION and the Myrdal-Yalkowsky method (Myrdal and Yalkowsky, 1997) are negligible. For these critical compounds, the vapor pressures estimated by SIMPOL.1 are in the range of other estimates except for APINBNO3 (the 5th

most abundant species in the aerosol phase) where it is an order of magnitude lower than the next highest estimate. Therefore, systematic underestimation of vapor pressure is not the obvious cause of overabundance of this product in our simulation.

### 3.2.2    Model-measurement comparison of FG mole fractions

Comparing with observations, discrepancies in the proportions of $CONO_2$ and naCO are higher than in the APIN-lNO$_x$ case. $CONO_2$ mole fraction is overestimated by the model as it accounts for 6% of the relative mole fraction after 20 hours, while

in the measurements it accounts for only 2% of the relative mole fraction (Figure 3). The model also over predicts the relative mole fraction of naCO (7% against less than 1% in the measurements).

The low relative humidity conditions of the experiments (RH <5%) exclude organonitrate hydrolysis (Liu et al., 2012), not included in the model, as a possible condensed phase pathway that explains the model-measurement discrepancy for $CONO_2$. Organonitrate compounds are formed from the addition of NO to a peroxy radical (e.g., C813NO3 is formed from the addition

of NO to C813O2). Yields are affected by the rate of $HO_2$ or $NO_3$ addition to the peroxy radical, and the branching ratio of the reaction to produce organonitrate or alkoxy radical and $NO_2$ (Noziere et al., 1999; Ruppert et al., 1999; Aschmann et al., 2002; Pinho et al., 2007). High uncertainty in $CONO_2$ production rates by lumped chemical reaction schemes has also been reported (Henderson et al., 2011), but uncertainties may also be present in explicit mechanisms for the reasons described. Smaller number of components condensing to the aerosol phase may lead to greater sensitivity of simulation results to individual

values of such rate constants or vapor pressures (which may otherwise be compensated across a larger suite of compounds or reactions), resulting in higher likelihood of discrepancies between model predictions and observations.





### 3.3 APIN-nNO$_x$

#### 3.3.1 Simulation results

The apportionment of O:C in the APIN-nNO$_x$ system is qualitatively similar to APIN-lNO$_x$, sans contributions from nitrogenated groups. The FG composition of gas and aerosol O:C ratios are very similar, though the value is higher in the latter
phase (Figure 1). The aerosol-phase O:C ratio increases in the simulation to arrive to 0.53 after 20 hours, while the observed O:C ratio by Chhabra et al. (2011) is between 0.3–0.4. We can see that in the very beginning of the simulations the only compounds contributing to the O:C ratio that are able to partition to the aerosol phase have aCOH and hydroperoxide moieties. The ketone FG starts contributing to O:C in the aerosol phase only after this initial phase. The hydroperoxide FG accounts for 42% of the total SOA O:C ratio after 20 hours of simulation.

As for APIN-hNO$_x$, the $C_{\mathrm{SOA}}$ formed in these simulations is an order of magnitude less than the 64 µg m$^{-3}$ reported for the corresponding experiment. Sensitivity analysis with respect to $C_{\mathrm{SOA}}$ (by varying the amount of absorptive mass) indicates that relative proportions analyzed are again representative for aerosol formed according to the chemical mechanism and vapor pressure estimation method in our simulation framework (Appendix B), which excludes accretion reactions in the condensed phase. The same four carbonyl compounds that make up 70% of the ketone CO comprise over 90% of the naCO, 80% of hydroperoxide, and 80% of the $C_{\mathrm{SOA}}$ in this system (Figure 7) at peak $C_{\mathrm{SOA}}$ (12.1 hours). In contrast to APIN-lNO$_x$ and APIN-hNO$_x$, multifunctional organonitrate compounds do not contribute to the COOH abundance; it is effectively accounted for by only two compounds: H3C25CCO2H (Table 2) contributing 60% and PINONIC (pinonic acid) contributing 40%. The reason for the large contribution of H3C25CCO2H is its saturation concentration of $\log_{10} C^0 = 1.2$, while PINONIC has a $\log_{10} C^0 = 3.0$ but its total (gas+aerosol) concentration is almost an order of magnitude more than H3C25CCO2H (Figure S13). As with the APIN-hNO$_x$ simulations, the dominance of so few polyfunctional compounds in the aerosol phase is surprising; past studies have identified more than five smaller compounds comprising observed APIN (and other precursor) ozonolysis aerosol yields under dry conditions (Yu et al., 1999; Pankow, 2001). These compounds are primarily composed of COOH, aCOH, and aldehyde CO groups, which are present in low abundance in our simulations.

#### 3.3.2 Model-measurement comparison of FG mole fractions

In Figure 3 we can observe that the highest discrepancies in the FG relative mole fraction between experimental observations and simulations are found in the oxygenated FGs (COOH, aCOH, and naCO). While the naCO mass fraction is overestimated by the model (7% in the model against 3% in the experiment), the COOH and the aCOH are underestimated (less than 1% in the model against 3% in the experiment for COOH and 7% against 15% for aCOH). Reactions of aldehydes with hydroperoxides can form peroxyhemiacetals (Jang and Kamens, 2001; Docherty et al., 2005), leading to a reduction in naCO. However, if the condensed-phase naCO is mostly ketone as predicted by the model (Figures 1 and S8), this is not likely to improve model-measurement agreement of the relative mole fractions of naCO.

Simulation of COOH production by gas-phase oxidation has also been reported to underestimate its abundance in OA in other studies (e.g., Aumont et al., 2012). In particular, there is a question whether the gas-phase production rate is low or





production and degradation rates are both high. To examine this question, IRR contributions to production and loss of COOH from semi-volatile compounds that can condense to the aerosol phase in appreciable proportions ($\log_{10} C^0 \leq 2.5$) are shown in Figure 8. The net change in moles of COOH due to degradation is 83% of the production for these compounds. There are more compounds contributing to the aerosol-phase COOH in the APIN-lNO$_x$ simulation and the net degradation is only 13% of the net production. One known mechanism for production of COOH by heterogeneous reactions not included in the model involves the transformation of hydroxy carbonyls (formed from alkoxy radicals) to dihydrofurans (Ziemann and Atkinson, 2012), which are further oxidized in the gas phase by O$_3$, primarily. However, as there is no NO$_x$ in the APIN-nNO$_x$ system, the O$_3$ production rate is small (concentrations are less than 0.6 ppb in our simulations, Figure S14). The model-measurement discrepancy may again be partially due to the low concentrations of condensible products and small number of products partitioning to the aerosol phase in this simulation, though additional oxidation mechanisms in the condensed phase may also contribute to this discrepancy.

### 3.4 TMB-lNO$_x$

#### 3.4.1 Simulation results

In the TMB-lNO$_x$ simulations, continued oxidation in the gas-phase proceeds for the entire duration of simulation and the O:C ratio approaches unity, while the aerosol-phase O:C and FG composition largely stabilizes in magnitude after the first several hours. In this mechanism, we note the presence of esters, ethers, organic peroxides (ROOR' + ROOH), and anhydrides which were not present in the APIN photooxidation schemes. The O:C ratio of the simulated SOA is ($\sim$0.7), exceeding values of 0.25--0.47 measured by Sato et al. (2012). The experiments of Sato et al. (2012) were also TMB photooxidation experiments with NO$_x$ and methyl nitrate as OH source, with no seed but lower RH ($<$1%) than Sax et al. (2005).

The peroxyacyl nitrate is the FG that contributes the most to the gas phase O:C ratio (around 30%), as in the APIN-lNO$_x$ case. Peroxide, hydroperoxide, and CONO$_2$ are major contributors to the aerosol O:C ratio and ester and aCOH are also present in the aerosol phase. COOH, anhydrides, peroxy acid, peroxyacyl nitrate and naCO contribute to the gas phase. The high peroxide and hydroperoxide contribution to the aerosol phase in this simulation agrees with their reported role in SOA formation from TMB photooxidation in low NO$_x$ conditions (Wyche et al., 2009). However, the $C_{\text{SOA}}$ mass fraction of compounds containing organic peroxide is 96% in our simulations, which is higher than what has been experimentally determined by Sato et al. (2012) in similar conditions (12±8%).

While higher precursor concentrations lead to high concentrations of condensible products in the gas-phase, two compounds (TM135BPOOH and NMXYFUOOH) make up over 70% of $C_{\text{SOA}}$ at its peak (11.9 hours). TM135BPOOH is found to contribute 60% to the aCH and 65% of hydroperoxide FGs in the aerosol phase at peak $C_{\text{SOA}}$ (Figure 9). TM135BPOOH is an hydroxy hydroperoxide bicyclic peroxide (Table 2) that is formed after many oxidation steps that follow the first addition of OH to the aromatic ring (Rickard et al., 2010). This compound has been also indicated by Rickard et al. (2010) as a potential SOA forming compound from TMB photooxidation and was found to be the most abundant hydroperoxide compound in the beginning of the photooxidation simulation they conducted.



### 3.4.2  Model-measurement comparison of FG mole fractions

The model is able to capture the general trends in COOH and $CONO_2$ relative to the first sample but the difference in magnitude is higher than APIN-lNO$_x$ case (Figure 2). The measured changes in COOH indicate an increase of a factor of four, while predicted COOH increases by as much as a factor of eight during the same time. The $CONO_2$ mole fraction at the end of

our simulation is around 0.7, while it arrives to 0.4 in the experimental observations. The increase in naCO and aCOH that are captured by simulation for APIN-lNO$_x$ are not captured for TMB-lNO$_x$. As for APIN-lNO$_x$, the degradation of the aCH fraction and the appearance of naCO and aCOH are slower in the model than in the experiments carried by Sax et al. (2005). From the molar fractions shown at different simulation times in Figure 3 we find better agreement in the beginning of the simulation and the differences in the FG molar fraction is higher compared to APIN-lNO$_x$.

The low proportion of naCO compared to simulations described in preceding sections is due to the lack of multifunctional ketone compounds. The ester CO abundance is on the same order as ketone CO (Figure S8) in our simulations, and it is possible that the naCO reported by Sax et al. (2005) may include ester CO as the absorption band at $\sim 1735$ cm$^{-1}$ is close to aldehyde and ketone CO at $\sim 1725$ and $\sim 1715$ cm$^{-1}$, respectively (Pavia et al., 2008). However, the ester CO contribution cannot explain the entire difference given the large discrepancy. Peroxide and hydroperoxide photolysis in the condensed phase

under UV irradiation can lead to the increase of both naCO and aCOH FGs, but increases calculated using the 6-day lifetime (Epstein et al., 2014) only partially explain this difference (with an increase from 7 to 8% for aCOH and from 1 to 3% for CO after 20 hours of irradiation). As there is indication that O:C and abundance of hydroperoxide and peroxide groups may be overestimated, it is possible that over-representation of TM135BPOOH in the simulated aerosol phase (Section 3.4.1) also contributes to an overestimation of aCH, leading to a smaller fraction of the measured oxygenated groups (COOH, aCOH, and

naCO).

## 4  Conclusions

In this study, the FG distribution of SOA generated in environmental control chamber experiments reported in literature for APIN and TMB photooxidation have been compared to explicit gas-phase chemistry and partitioning simulated with MCMv3.2 and SIMPOL.1. Varying degrees of agreement between the model and FTIR measurements of FG evolution in SOA generated

in environmental controlled chambers are found.

In the APIN-lNO$_x$ simulations, the FG relative abundance is well captured by the model in the first hours of simulation, and general trends in the changes of the mole fraction compared to the first sample are captured qualitatively by the model. However, the underestimation of the measured oxidized groups (COOH, aCOH, and CO) are apparent after 20 hours in our simulations; this discrepancy may be explained by heterogeneous reactions missing in the model. O:C is generally overestimated for APIN-

hNO$_x$, APIN-nNO$_x$, and TMB-lNO$_x$ on account of large contributions from $CONO_2$, peroxide, or hydroperoxide groups, while the aCH is simulated consistently in larger proportion to some of the measured oxygenated species (COOH and aCOH). These errors are largely correlated, as $C_{SOA}$ mass and individual FGs are dominated by a few polyfunctional compounds in these simulations. The dependencies of aerosol composition on a limited number of compounds also speaks to the sensitivity



of simulation results on a few kinetic or partitioning parameters, which might otherwise be averaged out in systems where the condensed phase is composed of a larger number of compounds.

In the APIN-hNO$_x$ simulations, the model predicts a higher fractional abundance of CONO$_2$ in the aerosol phase than what is observed in the FTIR measurements. The CONO$_2$ fraction arrives to constitute 46% of the total O:C ratio, which partly

contributes to the higher O:C of the aerosol phase during the simulation (0.78) compared to the O:C observed (∼0.4) by Chhabra et al. (2011). Only four CONO$_2$-containing polyfunctional compounds account for more than 80% of the organic mass. The uncertainties due to lack of kinetic data in the total CONO$_2$ yield in the primary oxidation sequence of APIN may play an important role in the high NO$_x$ regime and explain the discrepancies between model and measurements in this scenario. For the APIN-nNO$_x$ simulations, four polyfunctional compounds account for over 80% of the $C_{SOA}$ mass and a large bulk of

ketone CO and hydroperoxide FGs. The relative abundance of ketone CO is overestimated compared to observations; the O:C is also overestimated, possibly on account of the large (42%) contribution from the hydroperoxide FG which originates from the same set of molecules. For the TMB-lNO$_x$ photooxidation simulations, general trends in the changes in relative mole fractions compared to the first sample for COOH, aCOH, naCO, aCH and CONO$_2$ also qualitatively follow observations, but their magnitudes have more discrepancies with experiments than in the case of APIN-lNO$_x$. These discrepancies have also

been hypothesized as a sensitivity to reaction rates and vapor pressures of a few dominant products that contribute significantly to the aCH mole fraction and peroxide fraction of the aerosol O:C ratio. As for the APIN-lNO$_x$ simulations, the agreement in abundances of aCH relative to the measured set of oxidized FGs may also be explained by additional condensed phase oxidation chemistry not included in the model.

While we have uncovered only a fraction of the analysis capabilities that a FG perspective provides, we anticipate that the

tools and approaches introduced in this work can be used to aid further comparisons between model simulations of both gas and aerosol-phase chemistry in conjunction with emerging methods for FG quantification.

*Acknowledgements.* Funding was provided by the Swiss National Science Foundation (200021_143298). The authors also thank C. Dupuy for conducting initial tests during model development.





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





## Tables

**Table 1.** Summary of the experimental conditions studied in this work. For simplification, an ID has been given to each system.

| ID | Publication | Precursor | Measurement conditions |
|---|---|---|---|
| APIN-lNO$_x$ | Sax et al., 2005 | $\alpha$-pinene: 300 ppb | low NO$_x$ : 240 ppb <br> RH: 61% <br> seed: none <br> radical initiator: propene, 300 ppb |
| APIN-hNO$_x$ | Chhabra et al., 2011 | $\alpha$-pinene: 47 ppb reacted | high NO$_x$ : 847 ppb <br> RH: 5% <br> seed: ammonium sulfate, 27 µg m$^{-3}$ <br> radical initiator: CH$_3$ONO, 200-400 ppb |
| APIN-nNO$_x$ | Chhabra et al., 2011 | $\alpha$-pinene: 46 ppb reacted | no NO$_x$ <br> RH: 4% <br> seed: ammonium sulfate, 24 µg m$^{-3}$ <br> radical initiator: H$_2$O$_2$, |
| TMB-lNO$_x$ | Sax et al., 2005 | 1,3,5-trimethyl benzene: 1312 ppb | low NO$_x$ : 320 ppb <br> RH: 60% <br> seed: none <br> radical initiator: propene, 300 ppb |



**Table 2.** Illustration of several polyfunctional molecules discussed in Section 3.

| MCM name | Molecular weight | $\log_{10} C^0$ (298 K) | Structure |
|---|---|---|---|
| H3C25CCO2H | 174.1513 | 1.16 | |
| C719OOH | 176.1672 | 0.97 | |
| C97OOH | 188.2209 | 2.33 | |
| C98OOH | 204.2203 | 1.45 | |
| C106OOH | 216.2310 | 1.92 | |
| TM135BPOOH | 202.2045 | 2.78 | |
| NMXYFUOOH | 207.1382 | 3.40 | |
| C813NO3 | 235.1913 | -0.38 | |
| C811PAN | 247.2020 | 2.17 | |





**Table 3.** Comparison of pure component vapor pressures (atm) estimated (at 298 K) for the most abundant $CONO_2$ compounds in the aerosol phase for the APIN-hNO$_x$ simulation. Calculations were accessed using the UManSysProp tool (Topping et al., 2015)

| Compound | SIMPOL.1[1] | EVAPORATION[2] | Nannoolal[3] | Myrdal & Yalkowsky[4] |
|---|---|---|---|---|
| C813NO3 | $4.33 \times 10^{-11}$ | $4.04 \times 10^{-11}$ | $1.70 \times 10^{-11}$ | $7.05 \times 10^{-9}$ |
| C98NO3 | $6.12 \times 10^{-9}$ | $2.07 \times 10^{-8}$ | $8.15 \times 10^{-9}$ | $1.01 \times 10^{-7}$ |
| C719NO3 | $2.33 \times 10^{-9}$ | $3.34 \times 10^{-10}$ | $5.96 \times 10^{-9}$ | $3.45 \times 10^{-7}$ |
| APINANO3 | $1.55 \times 10^{-7}$ | $2.38 \times 10^{-6}$ | $9.47 \times 10^{-7}$ | $5.53 \times 10^{-6}$ |
| APINBNO3 | $1.55 \times 10^{-7}$ | $8.19 \times 10^{-6}$ | $1.45 \times 10^{-6}$ | $7.39 \times 10^{-6}$ |

[1]Pankow and Asher (2008)

[2]Compernolle et al. (2011)

[3]Nannoolal et al. (2008)

[4]Myrdal and Yalkowsky (1997)





**Figures**

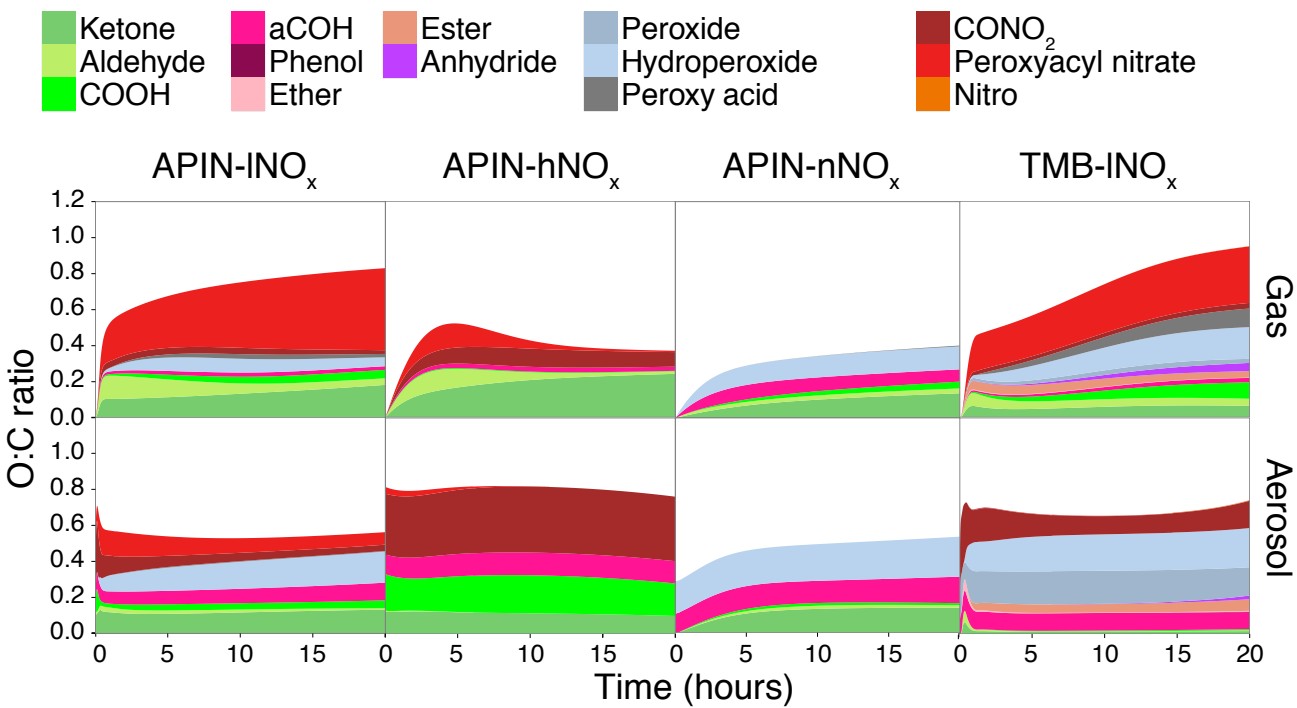

**Figure 1.** Time series of the relative molar contribution of different FGs to the O:C in the gas phase (top panel) and aerosol phase (bottom panel) simulated in this work for APIN-lNO$_x$, APIN-hNO$_x$, APIN-nNO$_x$, and TMB-lNO$_x$ cases. The contribution of each FG to the O:C ratio accounts for the number of oxygen atoms per FG.



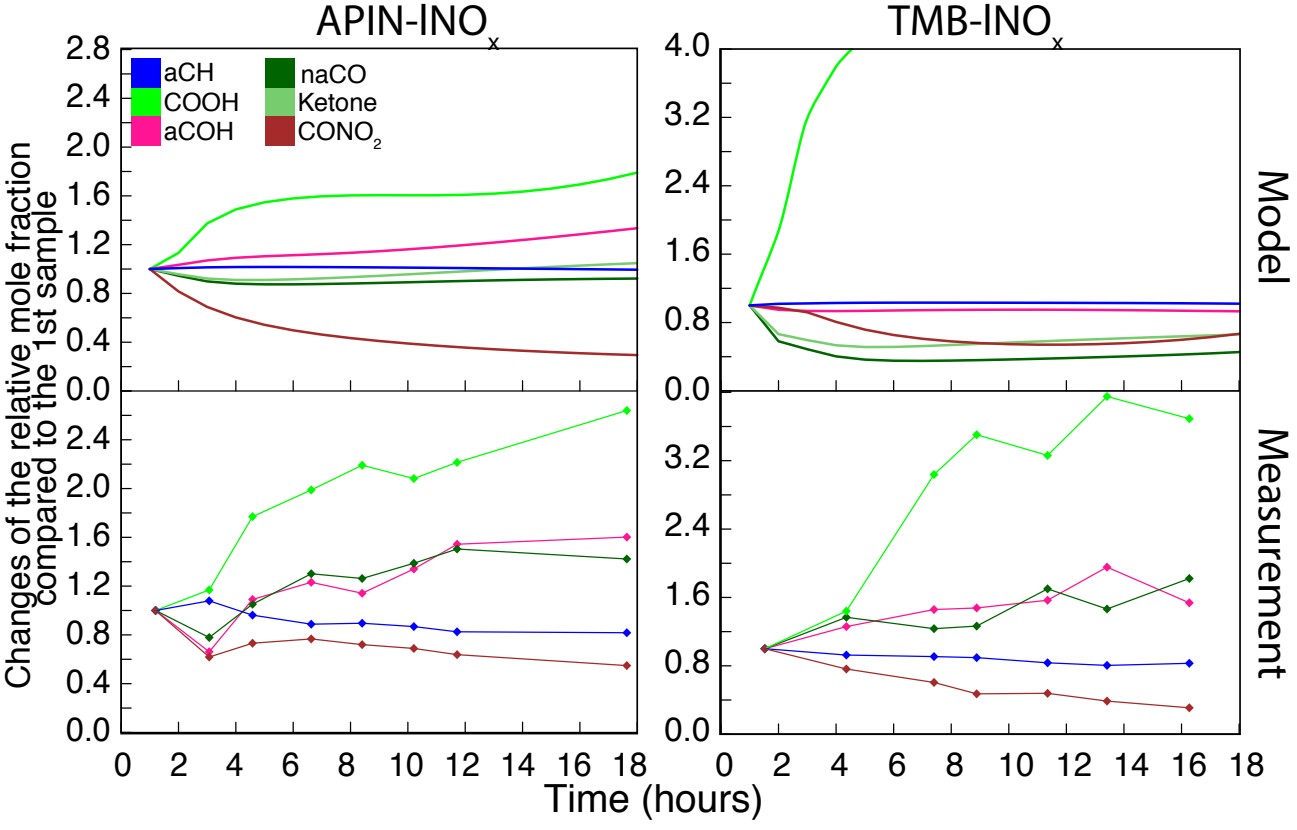

**Figure 2.** Comparison in the changes of the relative mole fraction compared to the first sample for COOH, COH, CO, aCH and $CONO_2$ of the aerosol phase measured by Sax et al. (2005) and modeled in this work for the APIN-lNO$_x$ and TMB-lNO$_x$ cases. The contribution of ketone and aldehyde to CO have been reported separately in the model results. The $x$-axis refers to the hours after the lights were turned on in the chamber for the top panel (Measurement), and the time after the start of the simulation in the bottom panel (Model).




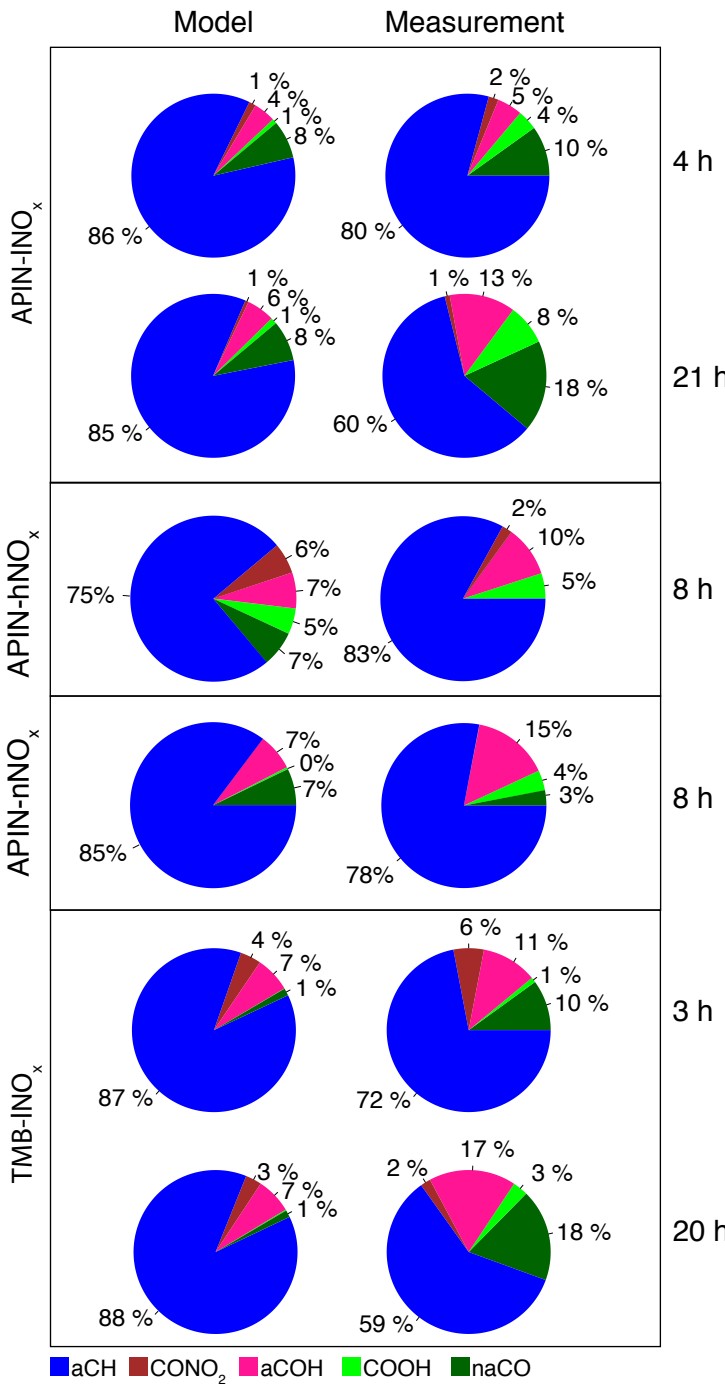

**Figure 3.** Pie charts illustrating the time-integrated relative aerosol mole fraction of aCH, CO, COOH, $CONO_2$, and aCOH in model simulations and experiments. The time reported refers to the hours after the lights were turned on in the chamber (Measurements), and the time after the start of the simulation (Model). In the pie charts reporting the measurement conducted by Chhabra et al. (2011) (APIN-hNO$_x$ and APIN-nNO$_x$) the $CNH_2$ fraction has been omitted in order to obtain a direct comparison between model and experiments. The sum of percentages combines to $100\pm1\%$, as individual values were rounded to the nearest whole number for labeling.





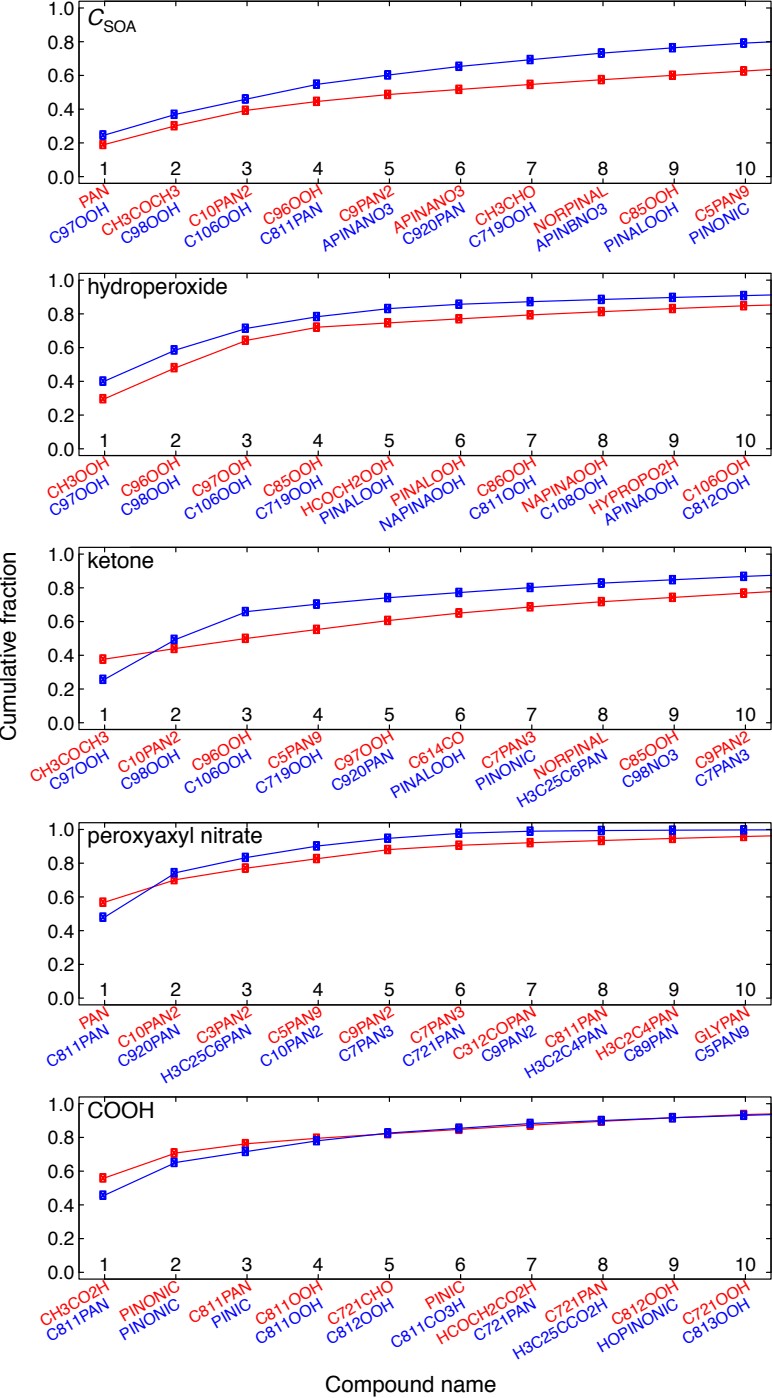

**Figure 4.** Cumulative contribution (as a fraction of total) of each compound to the overall $C_{SOA}$ mass and abundance of different FG fragments for the APIN-lNO$_x$ simulation. Compounds are arranged in order of decreasing contribution in each phase (i.e., first molecule contributes the greatest amount). Contributions to the aerosol phase are shown in blue and the gas phase in red.





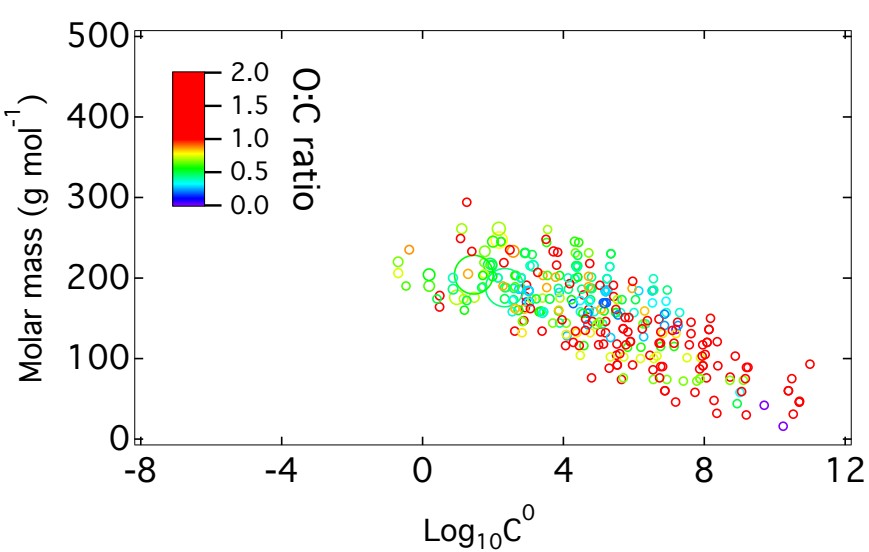

**Figure 5.** Molar mass vs the logarithm of the pure components saturation concentration for the compounds included in the APIN and propene MCMv3.2 degradation scheme. The size of the circles is proportional to the compound mass in the aerosol phase found in APIN-$lNO_x$ simulation.





**Figure 6.** Cumulative contribution (as a fraction of total) of each compound to the overall $C_{SOA}$ mass and abundance of different FG fragments for the APIN-hNO$_x$ simulation. Compounds are arranged in order of decreasing contribution (i.e., the first compound contributes most) for each phase. Contributions to the aerosol phase are shown in blue and the gas phase in red.





**Figure 7.** Cumulative contribution (as a fraction of total) of each compound to the overall $C_{SOA}$ mass and abundance of different FGs for the APIN-nNO$_x$ simulation. Componds are arranged in order of decreasing contribution (i.e., the first compound contributes most) for each phase. Contributions to the aerosol phase are shown in blue and the gas phase in red.





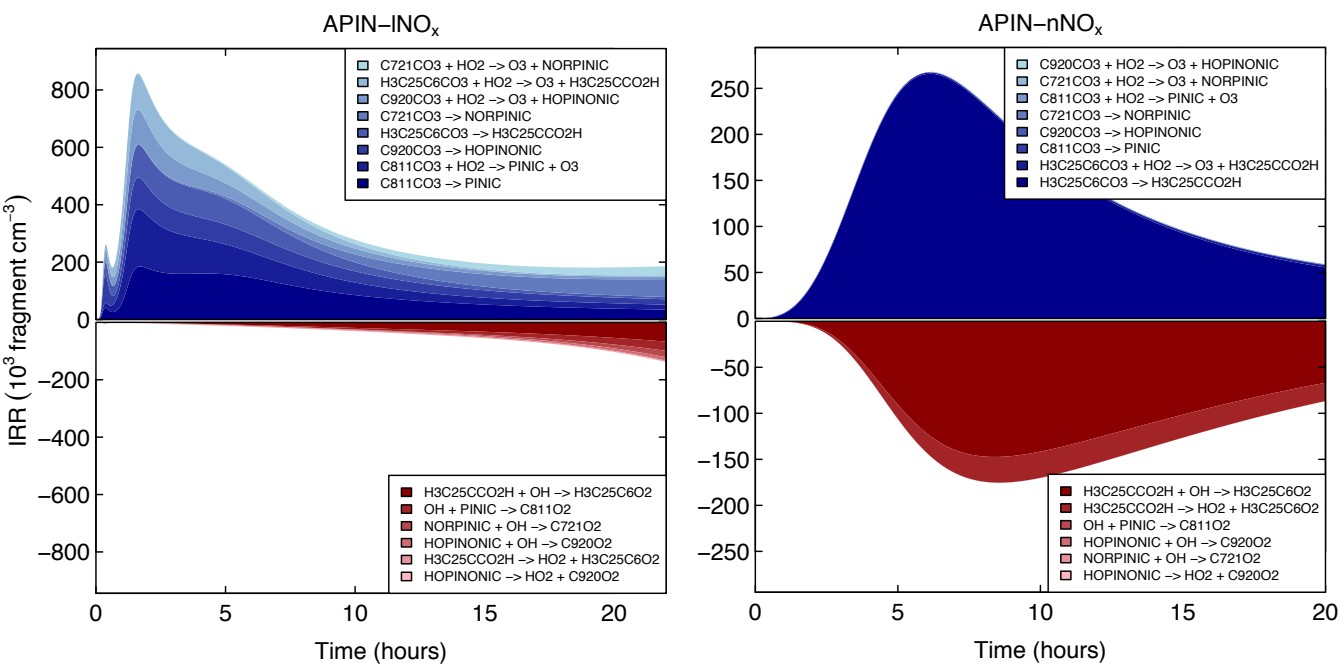

**Figure 8.** Integrated Reaction Rates for the COOH group (denoted in units of fragments per molecule).





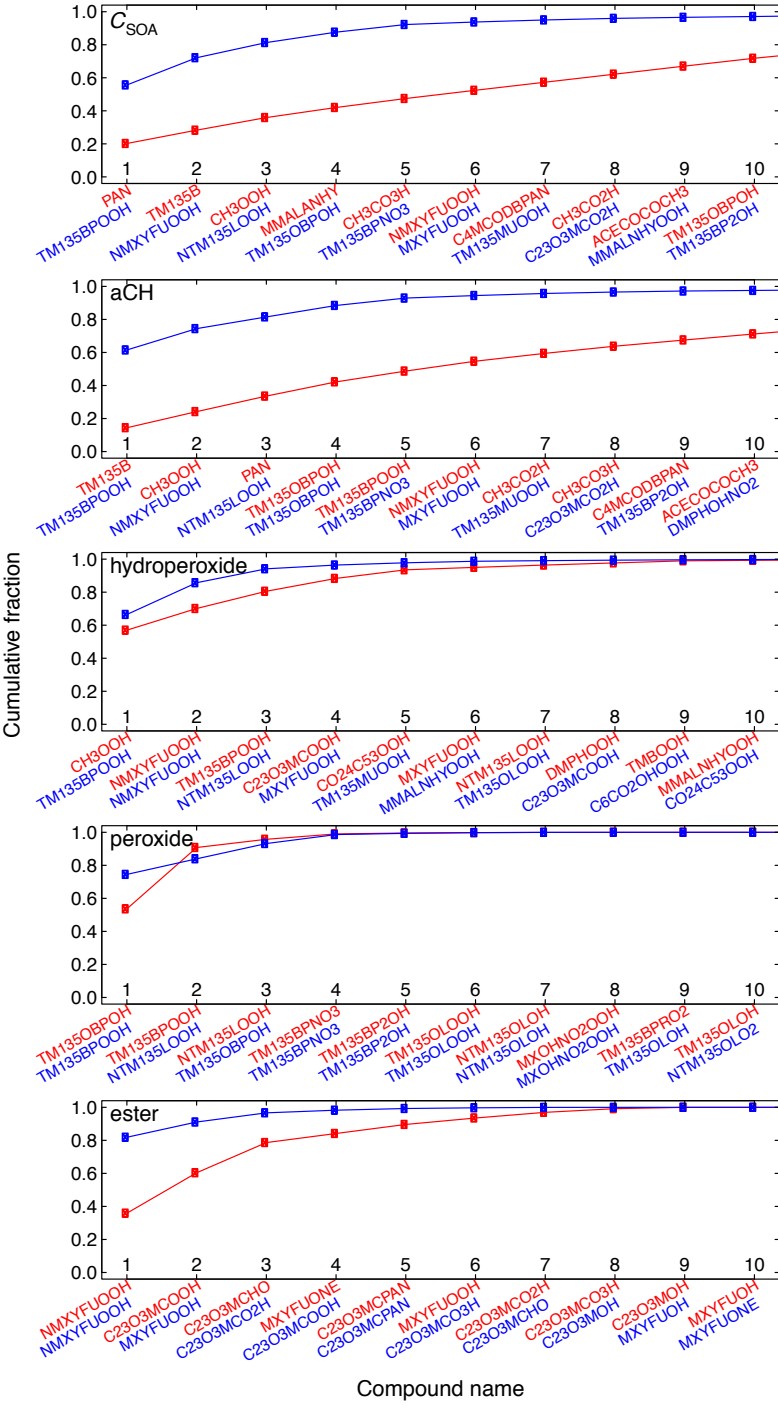

**Figure 9.** Cumulative contribution (as a fraction of total) of each compound to the overall $C_{SOA}$ mass and abundance of different FGs for the TMB-lNO$_x$ simulation. Componds are arranged in order of decreasing contribution (i.e., the first compound contributes most) for each phase. Contributions to the aerosol phase are shown in blue and the gas phase in red.