# Peer review of "Model-measurement comparison of functional group abundance in $\alpha$ -pinene and 1,3,5-trimethylbenzene secondary organic aerosol formation"

_Atmospheric Chemistry and Physics, 2016_

## Referee Comment (RC1) · Anonymous Referee #1 · 22 Mar 2016

General Comments

In this manuscript the authors compare results of Master Chemical Mechanism (MCM) simulations of the functional group composition of secondary organic aerosol (SOA) formed from reactions of a-pinene (APIN) and 1,3,5-trimethylbenzene (TMB) with measured values determined by FTIR analysis in order to demonstrate the utility of this approach for evaluating chemical reaction models. The systems chosen for study are popular ones since APIN is an important biogenic VOC and TMB is representative of anthropogenic aromatic emissions. The manuscript is concise and well written, and the authors do a good job of comparing model and measurement results, providing plausible explanations for discrepancies when possible. Because there are in many

cases significant differences, and both the simulations and measurements have considerable uncertainties, it is difficult to determine the source of the discrepancies. One conclusion is therefore that more measurements of chamber systems are needed using a variety of tools in order to develop databases of reliable chemical data for model comparisons. In general, however, I think the approach presented here has promise, and that the manuscript presents a useful demonstration of how models can be tested using more detailed chemical data rather than just SOA yields and O/C ratios. I think the paper is suitable for publication in ACP, but have a couple questions the authors should address.

Specific Comments

1. The comparison of measurements with simulations of the low-NOx TMB reaction seems problematic, since FTIR does not measure peroxides, which dominate the simulated SOA composition. I don't see any explicit mention of this.

2. When certain FG, like peroxides, are not measured by FTIR, how are the reported concentrations of the other FG affected? I do not see an "unidentified" component of SOA in the pie diagrams.

3. Is there molecular information available from other experiments conducted under roughly similar conditions that can be used to determine if the major molecular components predicted by the MCM are reasonable, which may help in determining whether the model or measurements are the source of discrepancies in some comparisons?

4. Are there simpler systems that might be modeled and analyzed?

Technical Comments

None.

---

## Referee Comment (RC2) · Anonymous Referee #3 · 5 Apr 2016

This manuscript introduces a modelling method than can be used for simulating secondary organic aerosol (SOA) formation and for studying especially functional group (FG) contributions in SOA. This kind of modelling approaches are necessary steps towards understanding SOA formation. The modelling framework is reasonable, manuscript is well written and the topic well within the scope of ACP. Therefore I recommend publication of the manuscript in ACP after minor revision.

Specific comments:

1) There are number of assumptions/estimations/simplification included in the model and the effect of these on the model results and for model-measurement comparison has not been discussed. Authors state that the relative abundances of functional

groups are robust with respect to many of the assumptions (P. 4, L. 23-24). However, I feel this statement needs to be justified. I recommend adding a short discussion/statement on the effect of the following:

a) Fixing radiation intensities in the model one constant value (P. 4, L. 6) Does this correspond to conditions in the experiments? If not, are the simulation results sensitive to this?

b) A seed is used for initializing the gas-particle partitioning (P. 4, L. 14-15). According to the appendix A the initial seed composition is calculated based on equilibrium gas-particle partitioning theory. Can this initial composition affect the model results at short simulation times? Also, I recommend adding here in the main text a sentence about how the initial seed composition was assigned.

c) The vapor wall losses where neglegted (P. 4, L. 20). Could this affect the simulations as the wall losses are likely volatility dependent and therefore may affect different FG differently?

P. 5, L. 24: 'FTIR analysis has been found to measure around 80 % of the organic mass...' How does the remaining 20 % affect the model-measurement comparison? Could the differences between modelled and measured FG relative contributions be explained in some cases by the missing 20 % of organic mass in measurements?

P. 8, L. 1-2: In addition to missing particle phase aggregation reactions, also highly oxidized high-molecular mass gas phase products might be missing from the model (Ehn et al. 2014, Nature 506, 476-479. Could such highly-oxidized organics explain the discrepancy between the model and measurement?

P. 13, L. 19-21: Here it would be good if the authors could summarize also that in which conditions their modelling tool may be applicable and what kind of improvements would be most important ones.

Authors mention that for APIN-hNOx and APIN-nNOx cases the model underestimates

the SOA mass by an order of magnitude. I was missing this information for the cases APIN-INOx and TMB-INOx.

Technical comments:

P. 7, L. 30-31: 'The sum of the oxidized fraction in the simulation. . .' This sentence is unclear, please revise.

P. 11, L. 14-26: Why are the model results here compared to Sato et al. measurements although the model simulations were designed to simulate the Sax et al. measurements?

Figure 2: Please clarify what 'relative mole fraction compared to the first sample' means. This can be interpreted at least as a ratio between molar fractions of compound i at time t and time t=0s or as moles of compound i at time t divided by moles of compound i at time 0 s.

Figure 2: Please clarify if the naCO includes both ketone and aldehyde for the model or if naCO includes only aldehyde for the model as ketone is presented also separately. At the moment it's not clear in measured naCO should be compared to modelled naCO or the sum of modelled naCO and ketone.

Figure 2: Consider plotting a vertical line at y = 1.0. That would make it easier to read the figure in case of those compounds that show only small changes.

---

## Author Comment (AC1) · 23 Jun 2016

**Response to Referee #1**

In this manuscript the authors compare results of Master Chemical Mechanism (MCM) simulations of the functional group composition of secondary organic aerosol (SOA) formed from reactions of a-pinene (APIN) and 1,3,5-trimethylbenzene (TMB) with measured values determined by FTIR analysis in order to demonstrate the utility of this approach for evaluating chemical reaction models. The systems chosen for study are popular ones since APIN is an important biogenic VOC and TMB is representative of anthropogenic aromatic emissions. The manuscript is concise and well written, and the authors do a good job of comparing model and measurement results, providing plausible explanations for discrepancies when possible. Because there are in many cases significant differences, and both the simulations and measurements have considerable uncertainties, it is difficult to determine the source of the discrepancies. One conclusion is therefore that more measurements of chamber systems are needed using a variety of tools in order to develop databases of reliable chemical data for model comparisons. In general, however, I think the approach presented here has promise, and that the manuscript presents a useful demonstration of how models can be tested using more detailed chemical data rather than just SOA yields and O/C ratios. I think the paper is suitable for publication in ACP, but have a couple questions the authors should address.

*We thank the reviewer for the support and helpful comments. We address specific comments below.*

1. **Comment** : The comparison of measurements with simulations of the low-NOx TMB reaction seems problematic, since FTIR does not measure peroxides, which dominate the simulated SOA composition. I don't see any explicit mention of this.

   **Response**: We note that peroxides concentrations were not reported in the study by Sax et al., but is not beyond the capability of analysis by FTIR as there are absorption bands in the infrared window for hydroperoxides and organic peroxides. However, the reviewer's point with respect to model-measurement comparison in this work is worth clarifying in the manuscript. In our pie charts in Figure 3, only the measured mole fraction of OA is shown.

   In Methods Section 2.4, we have added the sentence:

   "For model-measurement comparison, we select the subset of FGs that are reported by measurement and use relative metrics normalized only by the measured fraction of OA."

2. **Comment**: When certain FG, like peroxides, are not measured by FTIR, how are the reported concentrations of the other FG affected? I do not see an "unidentified" component of SOA in the pie diagrams.

   **Response** In response to the previous comment, we report relative concentrations of the measured fraction. While it is possible to include an unidentified or remaining fraction in the model simulation pie charts, we do not know how large this fraction should be for the FTIR measurements. We have therefore added in the caption of Figure 3 the phrase:

   "The mole fractions reported in simulations are summed with respect to the subset of FGs that are reported by measurement to facilitate direct comparison."

3. **Comment**: Is there molecular information available from other experiments conducted under roughly similar conditions that can be used to determine if the major molecular components predicted by the MCM are reasonable, which may help in determining whether the model or measurements are the source of discrepancies in some comparisons?

   **Response**: There are some analyses of gas phase composition which we now include in our analysis.

   In Results Section 3.1.1:

"Pinonic acid is the second largest contributor to COOH FG, which is consistent with previous reports of pinonic acid being a major contributor to SOA in APIN photooxidation over a range of $NO_x$ conditions Eddingsaas et al. (2012)."

And in Results Section 3.2.1: "As for APIN-lNO$_x$, pinonic acid is the second largest contributor to COOH FG; consistent with observations in similar experiments (Eddingsaas et al., 2012)."

We hope that this manuscript will encourage the adoption of FTIR as a complementary tool for OA analysis and that such joint measurements will become available in the future.

4. **Comment**: Are there simpler systems that might be modeled and analyzed?

**Response** $\alpha$-pinene dark ozonolysis is a classic, well-studied system (e.g., Yu et al., 1999) which may provide constraints and simplification in interpretation. At present time, we selected simulations for which experimental values were available, but hope that there will be opportunities for further exploration in future studies.

The reasoning for selecting these particular systems have now been explicitly added to the beginning of Methods Section 2:

"We target our model simulations to mimic SOA formation in environmentally controlled chambers for which FG measurements are available."

**References**

Eddingsaas, N. C., Loza, C. L., Yee, L. D., Seinfeld, J. H., and Wennberg, P. O.: alpha-pinene photooxidation under controlled chemical conditions - Part 1: Gas-phase composition in low- and high-NOx environments, Atmospheric Chemistry and Physics, 12, 6489–6504, doi:10.5194/acp-12-6489-2012, 2012.

Yu, J. Z., Cocker, D. R., Griffin, R. J., Flagan, R. C., and Seinfeld, J. H.: Gas-phase ozone oxidation of monoterpenes: Gaseous and particulate products, Journal of Atmospheric Chemistry, 34, 207–258, doi:10.1023/A:1006254930583, 1999.

---

## Author Comment (AC2) · 23 Jun 2016

**Response to Referee #3**

This manuscript introduces a modelling method than can be used for simulating secondary organic aerosol (SOA) formation and for studying especially functional group (FG) contributions in SOA. This kind of modelling approaches are necessary steps towards understanding SOA formation. The modelling framework is reasonable, manuscript is well written and the topic well within the scope of ACP. Therefore I recommend publication of the manuscript in ACP after minor revision.

*We thank the reviewer for the positive and helpful comments. We address specific comments below.*

1. **Comment**: There are number of assumptions/estimations/simplification included in the model and the effect of these on the model results and for model-measurement comparison has not been discussed. Authors state that the relative abundances of functional groups are robust with respect to many of the assumptions (P. 4, L. 23-24). However, I feel this statement needs to be justified. I recommend adding a short discussion/statement on the effect of the following:

    A. Fixing radiation intensities in the model one constant value (P. 4, L. 6) Does this correspond to conditions in the experiments? If not, are the simulation results sensitive to this?

    B. A seed is used for initializing the gas-particle partitioning (P. 4, L. 14-15). According to the appendix A the initial seed composition is calculated based on equilibrium gas-particle partitioning theory. Can this initial composition affect the model results at short simulation times? Also, I recommend adding here in the main text a sentence about how the initial seed composition was assigned.

    C. The vapor wall losses where neglected (P. 4, L. 20). Could this affect the simulations as the wall losses are likely volatility dependent and therefore may affect different FG differently?

   **Response**:

    A. The constant radiation intensities are indeed consistent with experimental conditions of Sax et al. (2005) and Chhabra et al. (2011).

       In the method section we have therefore added the phrase:

       "Radiation intensities were fixed at their maximum throughout the simulations to mimic conditions used in the chamber studies [...]."

    B. Based on the simulations varying seed composition in Appendix B, the sensitivity to initial seed composition is explored by varying the activity (mole fraction), $a_i = x_i$, according to initial seed concentration. We conclude that after an hour of simulation, relative FG abundances were generally invariant with initial seed composition.

       We have added the following statement to Methods Section 2.2:

       "We specify the bulk of $C_{\mathrm{OA,init}}$ to be a generic, non-volatile organic solvent that does not participate in reactions or partitioning, and is in equilibrium with the initial composition of the gas-phase (Appendix B). The relative composition reported in this study are insensitive to this value after one hour of simulation (Figures S3–S4)."

       We further note in Appendix B that this is in contrast to the approach of (Pankow, 2001):

       "Pankow (2001) used $C_{\mathrm{OA,init}}$ and composition ($x_i$s) from the final $C_{\mathrm{OA}}$ values determined from experiments. As this can introduce additional mass into the system, we introduce a different approach."

C. We appreciate the reviewers comment on this point. Under the assumption of a common wall loss parameter (Zhang et al., 2014), it is unlikely that our simulated composition would be altered even while the overall yield simulated would be decreased. However, wall losses that are composition dependent (Matsunaga and Ziemann, 2010; Yeh and Ziemann, 2015) may indeed reduce the most condensible substances in the system and alter the relative particle composition (Cappa et al., 2016; La et al., 2016). However, the magnitude of this effect also depends on the number of condensable species formed and their absolute concentrations. For instance, in the TMB-lNO$_x$ simulation, while gas and particle concentrations are observed to increase, the aerosol composition remains relatively constant. This is possibly due to the fact that the number condensed species is dominated by a few compounds that span a narrow range of saturation concentrations. Also, our model simulations tend to underpredict the oxygenated (polar) fraction in the condensed phase, indicating that the mechanisms for discrepancies suggested in the manuscript are likely to be even more active than current magnitude of differences suggest.

We have included the following comment in Methods Section 2.2:

"However, the impact of vapor losses to chamber walls may require investigation in future work. An assumption of a common wall loss parameter for all species (e.g., Zhang et al., 2014) would mostly reduce the overall yield from simulation, but compound-dependent wall losses (Matsunaga and Ziemann, 2010; Yeh and Ziemann, 2015) may preferentially reduce the concentration of the most condensible substances in the system and lead to a different relative particle composition (Cappa et al., 2016; La et al., 2016). The magnitude of this effect also depends on the number of condensable species formed, the range of saturation concentrations spanned, and their absolute abundance."

2. **Comment**: P. 5, L. 24: 'FTIR analysis has been found to measure around 80 % of the organic mass...' How does the remaining 20 % affect the model-measurement comparison? Could the differences between modelled and measured FG relative contributions be explained in some cases by the missing 20 % of organic mass in measurements?

**Response**: As the reviewer points out, not all FGs in the simulation that could be present in the physical system are reported by current FTIR calibration models, and this is a likely explanation for underestimation of OA (Russell, 2003). We have decided to remove the statement regarding the quantified fraction as this is more applicable for ambient samples, and is a topic of current study (alluded to in Methods Section 2.4: "Constraining the mapping of measured bonds to atoms for estimation of [mass contributions of functional groups] in various mixtures are planned for future work."). With regards to the model-measurement comparison, only relative fractions considering the measured FGs are shown to circumvent this potential ambiguity.

Also in response to Reviewer #1, We have added a phrase in the Methods Section 2.4 to clarify this point:

"For model-measurement comparison, we select the subset of FGs that are reported by measurement and use relative metrics normalized only by measured fractions of OA."

And in the caption of Figure 3:

"The mole fractions reported in simulations are taken with respect to the sum of FGs that are reported by measurement to facilitate direct comparison."

3. **Comment**: P. 8, L. 1-2: In addition to missing particle phase aggregation reactions, also highly oxidized high-molecular mass gas phase products might be missing from the model (Ehn et al. 2014, Nature 506, 476-479. Could such highly-oxidized organics explain the discrepancy between the model and measurement?

**Response**: We thank the reviewer for suggesting this hypothesis. Ehn et al. (2014) report a high mass fraction of (up to two-thirds) by ELVOCs at low to moderate ($\sim$10 $\mu$g m$^{-3}$) loadings. However, Zhang

et al. (2015) report that ELVOCs may play a smaller role with SVOCs constituting around 60% of the SOA mass at higher loadings, under which the APIN-lNOx and TMB-lNOx simulations fall (Figure S1). However, ELVOCs may play a significant role in the APIN-hNOx and APIN-nNOx scenarios where loadings are smaller. However, the hypothesized mechanism for the formation of ELVOCs are successive H-abstractions and $O_2$-additions to peroxy radicals. In the presence of $NO_x$, reactions of these radicals with NO may prevent significant ELVOC formation.

We have therefore inserted statements in each set of results to discuss the potential role of ELVOCs.

In Results Section 3.1.2:

"This observation suggests that the role played by the gas-phase production of polyfunctional, extremely low volatility compounds (ELVOCs) observed in greater abundance Ehn et al. (2014) at lower aerosol $C_{OA}$ loadings [...]"

In Results Section 3.2.1:

"While production of large, polyfunctional ELVOCs might be a prime candidate for explaining the mass discrepancy at these low $C_{OA}$ loadings (comprising up to two-thirds for mass concentrations less than 10 $\mu g\,m^{-3}$), reactions with NO with peroxy radicals may inhibit formation of ELVOCs through the hypothesized mechanism of H-abstraction and $O_2$-addition to peroxy radicals (Ehn et al., 2014)."

In Results Section 3.3.2:

"Production and condensation of ELVOCs or additional oxidation mechanisms in the condensed phase not implemented in our model may also contribute to this discrepancy."

In Results Section 3.4.1:

"While formation and condensation of ELVOCs in the experimental system cannot be ruled out, it is likely that their contribution would be much smaller than the SVOC fraction on account of the high mass loadings (Figure S1) (Zhang et al., 2015)."

4. **Comment**: P. 13, L. 19-21: Here it would be good if the authors could summarize also that in which conditions their modelling tool may be applicable and what kind of improvements would be most important ones.

    **Response**: We thank the reviewer for the suggestion to add a general summary.

    We have extended the Conclusion Section 4 with the following statements:

    "This work illustrates that concurrent measurement of FGs alongside common techniques for atomic and molecular characterization of OA can provide an opportunity for complementary evaluation and further guide detailed understanding of chemical and physical transformations. Analysis of FG abundance can supplement tracking of individual tracers and evaluate the importance of mechanisms that lead to production of a class of compounds in the overall molar (or mass) budget. FG abundances can also provide structural interpretation to variations in elemental ratios (e.g., O:C, H:C, and N:C). Looking forward, systematic model-measurement comparison of FGs under controlled conditions may be able to provide constraints and aid development of chemical mechanism generators (e.g., Gao et al., 2016; Aumont et al., 2005)."

5. **Comment**: Authors mention that for APIN-hNOx and APIN-nNOx cases the model underestimates the SOA mass by an order of magnitude. I was missing this information for the cases APIN-lNOx and TMB-lNOx.

    **Response**: The mass yields were not reported in the APIN-lNO$_x$ and TMB-lNO$_x$ studies as the stated objective of Sax et al. (2005) was to "monitor the temporal change of the functional groups present in the SOA."

Technical comments:

6. **Comment**: P. 7, L. 30-31: 'The sum of the oxidized fraction in the simulation...' This sentence is unclear, please revise.

   **Response**: We have modified this sentence to read: "The oxidized fraction in the simulation is consistently lower than in the measurements, as it remains below 16% in the model while it increases to 40% after 20 hours in the reported measurements."

7. **Comment**: P. 11, L. 14-26: Why are the model results here compared to Sato et al. measurements although the model simulations were designed to simulate the Sax et al. measurements?

   **Response**: Our simulation is based on Sax et al. (2005) to target comparison of FG measurements. But as Sax et al. (2005) do not report overall O:C ratios in their study, we provide a comparison of simulation results to the overall O:C ratio reported by Sato et al. (2012) for experiments conducted under similar conditions.

   We added a modified the phrase to clarify this point:

   "The O:C ratio of the simulated SOA is ($\sim$0.7). While overall O:C in experiments of Sax et al. (2005) were not reported, Sato et al. (2012) report values in the range of 0.25–0.47 for similar TMB photooxidation experiments with $NO_x$ and methyl nitrate as OH source, with no seed and lower RH (<1%)."

8. **Comment**: Figure 2: Please clarify what 'relative mole fraction compared to the first sample' means. This can be interpreted at least as a ratio between molar fractions of compound i at time t and time t=0s or as moles of compound i at time t divided by moles of compound i at time 0 s.

   **Response**: This is a metric and figure format we adopted from Sax et al. (2005). For clarification, we have added the following statement in the caption of Figure 2: "For a chosen FG the changes of the relative mole fraction compared to the first sample is calculated as the ratio between the relative mole fraction at the chosen time and the relative mole fraction at 1 hour."

9. **Comment**: Figure 2: Please clarify if the naCO includes both ketone and aldehyde for the model or if naCO includes only aldehyde for the model as ketone is presented also separately. At the moment it's not clear in measured naCO should be compared to modelled naCO or the sum of modelled naCO and ketone.

   **Response**: We thank the reviewer for this opportunity for clarification. In the Figure 2 caption, we now write: "naCO includes ketone and aldehyde FGs, but the change in relative ketone FG abundance is also shown separately for illustration."

10. **Comment**: Figure 2: Consider plotting a vertical line at [y] = 1.0. That would make it easier to read the figure in case of those compounds that show only small changes.

    **Response**: The dashed line has been added to the figure and we have added also a phrase in the caption: "The dashed line corresponds to y=1 and has been added for visual reference."

**References**

[revised manuscript text omitted]